# Factors Influencing Biofilm Formation by *Salmonella enterica* sv. Typhimurium, *E. cloacae, E. hormaechei*, *Pantoea* spp., and *Bacillus* spp. Isolated from Human Milk Determined by PCA Analysis

**DOI:** 10.3390/foods11233862

**Published:** 2022-11-30

**Authors:** Mateusz Gemba, Elżbieta Rosiak, Zuzanna Nowak-Życzyńska, Paulina Kałęcka, Elżbieta Łodykowska, Danuta Kołożyn-Krajewska

**Affiliations:** 1Institute of Human Nutrition, Department of Food Hygiene and Quality Management, Warsaw University of Life Sciences—SGGW, 02-786 Warsaw, Poland; 2Institue of Animal Sciences, Departament of Food Hygiene and Quality Management, Warsaw University of Life Sciences—SGGW, 02-768 Warsaw, Poland; 3St. Holy Family Specialist Hospital, Regional Bank of Human Milk, 02-544 Warsaw, Poland

**Keywords:** bacterial biofilm, pathogen, breast milk, polystyrene, PCA

## Abstract

Bacteria enter milk during poor hygiene practices and can form a biofilm on surfaces that come into contact with human milk. The presence of a biofilm increases the risk of infections among newborns as bacteria protected by biofilm are resistant to washing and disinfection processes. The formation of the biofilm depends on the microbial species, environmental conditions, and the specific materials colonized. The aim of this study is to analyze the effects of factors such as temperature, incubation time, and initial cell concentration on biofilm formation by pathogenic bacteria isolated from human milk on model hydrophobic polystyrene surfaces. Model studies confirm that pathogenic bacteria appearing in human milk as a result of cross-contamination tend to form a biofilm. The majority of isolates formed biofilm at both 25 and 37 °C after 12 h at 1 × 10^3^ CFU/mL inoculum count. Multivariate principal component analysis (PCA) showed that at lower temperatures, biofilm formation by bacterial isolates was the main determinant of biofilm formation, other factors were less important; however, at 37 °C, time was a factor in biofilm formation. The model research performed underlines the importance of maintaining the proper hygiene of rooms, surfaces, and devices for expressing, storing, and preparing mothers’ milk and powdered infant formula (PIF) in facilities responsible for feeding newborns and premature babies.

## 1. Introduction

The presence of microflora in human milk is a result of the colonization of various organs, including the milk ducts, by select microorganisms. Sources of microflora include the transfer of the mother’s intestinal bacteria via the intestinal-glandular route, as well as the transfer of the mother’s skin microflora or the infant’s mouth microflora during suckling [1,2]. Commensal microflora identified in human milk includes *Streptococcus*, *Staphylococcus, Serratia, Pseudomonas, Corynebacterium, Ralstonia, Propionibacterium, Sphingomonas, and Bradyrhizobiaceae* spp. These bacteria were isolated from all human milk samples, independent of the donor. In addition, the presence of the following microorganisms has been confirmed in human milk: *Lactobacillus, Ruminococcus, Bacteroides, Bifidobacterium, Coprococcus, Faecalibacterium,* and *Roseburia* spp. [1,2].

Inappropriate hygiene during the handling of human milk and powdered infant formula (PIF) can result in infections in premature and newborn babies, and several cases of neonatal infections after human milk consumption have been reported. At least one infant was reported to have died in Australia (2015) and the USA (2016) as a result of a *Cronobacter sakazakii* infection contracted from a breast pump. Both cases indicate that poor hygiene practices (such as insufficient washing and disinfection, reusing of utensils to prepare and store PIF and human milk, or non-sterile storage of human milk) can result in cross-contamination of human milk and transmission of pathogens to the infant [3,4,5]. Cases of neonatal infection with *Klebsiella pneumoniae*, *Bacillus cereus,* and *Enterobacter cloacae*, due to ingestion of infected human milk, have also been reported in the literature [6,7,8,9]. At a Food and Agriculture Organization and a World Health Organization expert meeting in 2004, pathogenic bacteria associated with powdered infant formula were divided into three categories: category A, high causal relationship (*C. sakazakii* and *Salmonella enterica*); category B, a causal relationship is probable but not yet confirmed (*Pantoea agglomerans*, *Escherichia coli*, *Hafnia alvei*, *Citrobacter koseri*, *C. freundii*, *Klebsiella pneumoniae*, *K. oxytoca*, and *E. cloacae*); category C, a causal relationship less probable or not yet demonstrated (*Bacillus cereus*, *Clostridium difficile*, *C. perfringens*, *C. botulinum*, *Staphylococcus aureus,* and *Listeria monocytogenes*) [10].

Many species and genera of the following bacteria have demonstrated the ability to produce biofilm on usable surfaces: *E. coli*, *C. sakazakii*, *P. aeruginosa*, *S. aureus*, *S. epidermidis*, *Proteus mirabilis*, *K. pneumoniae*, *Enterococcus faecalis*, *L. monocytogenes*, *Streptococcus* spp., *Serratia* spp., and *Salmonella* spp. [11,12,13,14,15,16,17,18,19]. The biofilm adversely affects the ability to sanitize usable surfaces and can lead to cross-contamination of products. Cleaning and disinfectant treatments do not always prevent biofilm formation because disinfectants that are effective at removing plankton cells might not always remove biofilms [20,21]. The biofilm provides a physical barrier protecting bacteria from adverse environmental conditions. Biofilm-forming microorganisms show resistance to desiccation, the presence of antibiotics, heating, anaerobic conditions, and varying pH. Bacterial biofilm formation is a multi-step process involving four main phases: reversible adhesion, irreversible adhesion, maturation, and dispersion. The formation of biofilm depends on the microbial species, environmental conditions, and the specific materials colonized [22,23,24,25,26,27,28]. The first two phases of biofilm formation depend on the interaction of bacterial cell structure with the physico-chemical properties of the medium (such as hydrophobicity and electrostatic charge). Electrostatic, covalent carbon–carbon and hydrogen bonds are responsible for the adhesion process to the medium surface. These interactions allow bacterial cells to colonize surfaces such as plastics, metals, and glass [29]. Flagella, fimbriae, and curli are bacterial organelles that enable adhesion and biofilm formation. Flagella can affect the adhesion and biofilm formation of *E. coli*, *L. monocytogenes*, *Yersinia enterocolitica*, and *P. fluorescens*, while Type 1 fimbriae are responsible for initial bacterial adhesion in *E. coli*, *S. enterica* serovar *Enteritidis,* and *K. pneumoniae* and Type 3 fimbriae facilitate adherence of *K. pneumoniae* to glass. The presence of curli increases the ability of some strains, such as *E. coli*, to produce biofilm [17,30,31,32].

Temperature is one of the most important environmental factors in biofilm formation, which is most commonly observed in the temperature range of 20–37 °C. Higher temperatures increase the bacterial growth rate and glycocalyx synthesis, which also affects the rate of biofilm production. However, there is greater curli expansion at 30 °C than at 37 °C, providing greater bacterial adhesion to the plastic surface. Microbial adhesion is also affected by the growth phase of the microorganisms, in that cells in the logarithmic growth phase adhere faster than cells in the stationary phase. Biofilm production is also affected by nutrient availability in the environment and low nutrient availability (such as an absence of glucose) has been reported to promote biofilm formation. The presence of phosphorus increases the hydrophobicity of bacterial cells and enhances their biofilm-forming capacity. The ability to form a biofilm layer is also enhanced by the presence of other microorganisms and some bacterial cells in pure cultures do not adhere [27,28,33,34,35,36,37].

In summary, bacteria enter milk during poor hygiene practices and can form a biofilm on surfaces that come into contact with human milk. Contamination of human milk by pathogenic bacteria can cause neonatal disease and the presence of a biofilm increases the risk of infections among newborns, as bacteria protected by biofilm are resistant to washing and disinfection processes. Biofilm formation is a complex process that depends on many environmental conditions. The aim of this study is to analyze the effect of factors such as temperature, incubation time, and initial cell concentration on biofilm formation by pathogenic bacteria isolated from human milk on model hydrophobic polystyrene surfaces.

## 2. Materials and Methods

### 2.1. Material

Twelve human milk isolates, capable of growing on media and suitable for detection of *Cronobacter* spp. and referee strain *Cronobacter sakazakii* ATCC 29544, were used.

### 2.2. Bacterial Isolates

The bacterial isolates were detected in human milk samples according to the procedure described in ISO 22964: 2017 (Microbiology of the food chain: horizontal method for the detection of *Cronobacter* spp.). DNA isolation of presumptive isolates was performed according to the manufacturer’s instructions using the Extreme DNA Bacteria Kit (Blirt, Gdańsk, Poland).

### 2.3. Genus and Species Identification

Extracts of each strain were used to obtain sequences encoding 16S rRNA mitochondrial DNA (mtDNA). A fragment of mitochondrial DNA approximately 1400 nucleotides long was amplified using primers described by Berthold-Pluta et al. (2017) [38]. Amplification conditions were as follows: Initial denaturation for 4 min at 94 °C, followed by 35 cycles of 30 s at 94 °C, 30 s at 60 °C, and 45 s at 72 °C. The final extension step consisted of 15 min at 72 °C. The final amplification volume of 50 μL contained 25 μL of Perpetual OptiTaq PCR Master Mix (2x) (EurX, Gdańsk, Poland) and 2.0 μM of each primer. The clean-up reaction of PCR products before sequencing was performed using a PCR/DNA Clean-Up Purification Kit (EurX, Poland), in line with the manufacturer’s protocol. The sequencing of PCR products was performed using a BigDye™ Terminator v3.1 Cycle Sequencing Kit (Thermo Fisher Scientific, Waltham, MA, USA) in a volume of 10 μL, with purification entailing a BigDye Xterminator™ Purification Kit (Thermo Fisher Scientific). The electrophoresis was carried out in an ABI 3500 Genetic Analyzer (Applied Biosystems). DNA sequence chromatograms were analyzed using Mega 7.0.21 [39]. The fragments were sequenced twice with both light- and heavy-stranded primers. The sequences were aligned using BioEdit software v.7.0.5.3 (Manchester, UK) and alignments were checked manually. Finally, the consensus sequences of 16s rRNA were compared with NCBI database, using nucleotide BLAST.

### 2.4. Variants of Experiments

Overnight cultures of bacterial isolates were incubated in brain heart infusion broth (BHI) (Neogen, Warsaw, Poland) at 37.0 °C. Target inoculum concentrations of the cultures were reached by diluting the overnight culture up to 10^−7^ CFU/mL/mL and obtaining the following densities of bacterial cells: small (S), 1 × 10^2^ CFU/mL; medium (M), 1 × 10^3^ CFU/mL; large (L), 1 × 10^6^ CFU/mL. The inoculum density was checked by surface plating by dilutions in the range of 10^−5^–10^−7^. The sixty variations of conditions influencing biofilm production were carried out for each isolate. Each set of variables in the experiment was coded according to the following scheme: inoculum count (S, M, L)_time [h] (8, 12, 24, 48, 96) temperature [°C] (4.0, 25.0, 37.0, 62.5) (Figure 1).

### 2.5. Surface Material and Preparation

Biofilm production was tested on polystyrene surfaces (Nest Scientific Biotechnology, Wuxi, China). Sterile polystyrene 96-well (type F) microplates were used for adherent culture. The following temperatures were applied: 62.5 °C (temperature for Holder Pasteurization of human milk); 37.0 °C (human body temperature); 25.0 °C; 4.0 °C (typical temperature for human milk storage in room/cooling conditions).

### 2.6. Biofilm Formation on Hydrophobic Material (Polystyrene)

For biofilm formation in microplates, the bacterial strains were grown on BHI broth as described above. Three wells of sterile 96-well polystyrene plates were filled with 200 µL of bacterial suspension. Negative control wells contained BHI broth only. The plates were covered and incubated for 8, 12, 18, 24, 48, and 96 h (only in the case of incubation at a temperature of 4.0 °C was the time extended to 96 h). Additionally, 20 µL of bacterial suspension were transferred to three wells of another sterile 96-well polystyrene plate containing 180 µL of BHI broth and incubated at 4.0, 25.0, 37.0, and 62.5 °C for 24 h to assess the ability of the isolates to grow. The cell concentration was determined spectrophotometrically by measuring the absorbance at 600 nm in relation to the blank (BHI broth). The content of each plate was then removed, rinsed six times with water, and air-dried for 45 min. Once dry, the plates were stained with 200 μL of 1.0% crystal violet (Analab, Warsaw, Poland) for 30 min. Then, the content of each plate was removed, rinsed six times with running tap water, and air-dried for 45 min. The crystal violet bound to the biofilm was dissolved with 200 μL of 96% ethanol (Poch, Gliwice, Poland) for 10 min. From each well, 150 μL was added to a new sterile 96-well polystyrene plate, and absorbance was determined at 600 nm by a microplate reader (SoftMax Pro, San Jose, CA, USA). The cut-off optical density (ODc) for the plate test was defined as three standard deviations above the mean OD of the negative control and strains were classified as follows: non-adherent (−) OD600 < ODc; weak (+) ODc < OD600 < 2 × ODc; moderate (++) 2 × ODc < OD600 < 4 × ODc; strong (+++) 4 × ODc < OD600. The assay was repeated in triplicate, in two separate experiments for each strain, and the results were averaged.

In order to determine the differences in the degree of biofilm formation between bacterial isolates in response to temperature, inoculum size, and time, the Pearson Chi2 test was used. The Spearman rank test was used to assess the strength of the relationship between the variables. The differences in biofilm formation were considered statistically significant at *p* ≤ 0.05 for all tests. Statistica 13.3 PL software (TIBCO Software Inc., San Francisco, CA USA was used to perform the multidimensional principal component analysis (PCA) to select the surface-inoculum count-time-temperature variant most favorable for biofilm formation and to determine the correlation coefficient between components and inoculum count-time-temperature variants.

## 3. Results

### 3.1. Identification of Bacterial Isolates of Humane Milk

Based on the sequencing of the 16s rRNA, the following genera and species of isolates were identified: *Klebsiella* spp. (no. 1–5), *Bacillus* spp. (no. 1) *Pantoea* spp. (no. 1), *S. enterica* subsp. enterica serovar *Typhimurium* (no. 1), *Enterobacter hormaechei* (no. 1 and 2), and *E. cloacae* (no. 1 and 2).

### 3.2. Evaluation of Biofilm Formation on a Hydrophobic Surface

On the hydrophobic polystyrene surface, isolates of bacterial strains did not form biofilm at 4.0 °C and 62.5 °C during the experimental period of 8–96 h (Table 1). At 25.0 °C, *Pantoea* spp., *S. enterica* sv. *Typhimurium,* and *E. cloacae* 1 did not form biofilm at 8–48 h. After 8 h of incubation at 25.0 °C, none of the isolates nor the reference strain *C. sakazakii* ATCC 29544 formed biofilm, regardless of the inoculum concentration. Isolates of *Klebsiella* spp. no. 2 and 4 also did not form biofilm after 12 h of incubation. A strong biofilm was formed at this temperature by *Bacillus* spp. (after 12 h of incubation), by *E. cloacae* no. 2 (regardless of inoculum concentration), and by *Klebsiella* spp. 1 (S and M inoculum size). After 48 h of incubation, only the *E. cloacae* strain 2 formed a very strong biofilm at 25.0 °C.

At 37.0 °C, *Bacillus* spp., *E. cloacae* no. 2, and *Klebsiella* spp. no. 1, no. 3, and no. 5 formed the strongest biofilms, regardless of inoculum concentration. It was observed that isolates that did not produce a biofilm (*Pantoea* spp., *S. enterica* sv. *Typhimurium,* and *E. cloacae* no. 1) or produced weak biofilm (*Klebsiella* spp. 2) at 25.0 °C also produced a weaker biofilm at 37.0 °C (Table 1). 

At 37.0 °C, two strains showed a reduction in biofilm formation, while three strains at 37.0 °C produced more biofilm. The reference strain *C. sakazakii* formed a medium biofilm at 25.0 °C after 24 h of incubation at medium and high inoculum counts, while at 37.0 °C it formed a moderate biofilm after 12 h regardless of inoculum concentration. Under other conditions, it formed weak or no biofilm. Strong biofilm was formed by the other tested bacterial strains after 12 h of incubation.

At 25.0 °C and after 8 h of incubation, none of the analyzed bacterial isolates nor the *C. sakazakii* strain formed biofilm on a polystyrene surface. At low inoculum counts (S), no biofilm formation was observed in 58% of the isolates at up to 48 h. Medium and large inoculum (M and L) increased the percentage of strains forming biofilm within 24 h of incubation, while after 48 h of incubation, 75% and 83% of strains, respectively, no longer formed biofilm. Most of the isolates formed a strong biofilm after 12 h of incubation at an M inoculum count (Table 2).

Moderate biofilm was produced by isolates after 24 h of incubation with M inoculum count, while weak biofilms of isolates were most prolific after 24 h with S and L inoculum. At 25.0 °C, the classification of biofilm strength was not significantly influenced by inoculum density. There was also no correlation between inoculum density and biofilm strength at the time of its formation (Table 2).

At 37.0 °C, after 8 h of incubation (as opposed to 25.0 °C where bacteria did not form a biofilm), biofilm was formed by 50–58% of the tested bacteria and the percentage of strains not producing biofilm decreased during the 8–48 h incubation period. This indicates a statistically significant influence of temperature and individual characteristics of the tested species of bacteria on biofilm formation within 12–48 h (Table 3, *p* ≤ 0.05). After 12 h of incubation, the tested bacteria formed the strongest biofilm, regardless of the inoculum density. The observed differences in biofilm classification after 12 h of incubation were significantly different (Table 2). There was a tendency for weak biofilm formation after the longer incubation times of 24 and 48 h in the cases of the S and L inoculation size. In contrast, the bacterial inoculum of M-concentration favored the production of weak biofilm after 8 h.

### 3.3. Analysis of Factors Determining Biofilm Formation

A multidimensional principal component analysis is one of several techniques used to reduce the multidimensionality of variables. In the experiments conducted for the purpose of this study, each observation was the result of the influence of several variables (incubation time, inoculum size, and temperature). PCA is a statistical procedure to reduce the number of variables, detect, and verify structure and regularity in the relationships between variables. In the case of 53.8% of the tested bacterial isolates, the variables selected for the experiment explained 100% of the biofilm formation. For the remaining 46.2% of isolates, variability in biofilm formation is determined by two principal components (Figure 2). The strength of the compound between the components and the variables is expressed by the values of the correlation coefficients (Figure 2). The orthogonal axes of the graphs show the principal components determining biofilm formation by the tested bacterial strains and the *C. sakazakii* ATCC 29544 reference strain. The correlation circle determines the maximum value of the correlation between the variables and components 1 and 2. The points marked on the graph correspond to the factor loadings (correlations) between the variables and the principal components. The length of the directional vectors connecting the points corresponding to the variables with the origin of the coordinate system is proportional to the strength of the correlation of the variable with the component. All the obtained results take correlations (factor loadings) close to unity, so the points are very close to the unit circle, with the exception of three variable systems (S_8h, S_24h, and L_24h), for which the first two components explained 82, 78, and 80%, respectively, of the variation in biofilm formation by *S. enterica* sv. *Typhimurium* at 37.0 °C (Table 4 and Figure 2).

The angle between the vectors reflects the similarity of the effect of the variables on components 1 and 2 and the strength of the compound between the variables (Figure 2). The extracted component number 1 largely explained the conditions of biofilm formation by the tested bacterial strains (*C. sakazakii, Bacillus* spp., and *S. enterica* sv. *Typhimurium*). Two-dimensional planes spanning two principal component axes explained 100% of the variability in response to 24 sets of variables, representing conditions affecting biofilm formation by the 11 bacterial isolates tested and the reference strain (Table 4). In the case of *S. enterica* sv. *Typhimurium*, three components were isolated, explaining 78.09%, 15.95%, and 5.95% of the variation in biofilm formation, respectively. The number of principal components was reduced and the first two components were accepted to explain 94.04% of the conditions for biofilm formation by *S. enterica* sv. *Typhimurium*.

## 4. Discussion

Human milk may be contaminated with *Klebsiella* spp., *Bacillus* spp., *Pantoea* spp., *S. enterica* subsp. enterica serowar *Typhimurium*, *E. hormaechei*, or *E. cloacae*; these bacteria can infect newborns. The literature reports cases of infection in newborns with *K. pneumoniae*, *B. cereus,* and *E. cloacae* bacteria caused by the consumption of contaminated human milk [6,7,8,9,40,41].

The isolated bacteria have the ability to produce a biofilm. This study looked at the factors contributing to biofilm formation. Time, temperature, type, and species of bacteria as well as inoculum concentration were analyzed. At temperatures 4 and 62.5 °C, the isolates did not produce biofilm; however, most of the isolates showed growth at 62.5 °C after 8 h of incubation. Fakruddin et al. (2014) showed that the D_62_ value for *C. sakazakii* isolates ranged from 0.57 to 1.12 min [42]. Morgan et al. (1988) determined the predicted D_60_ value for *K. pneumoniae* isolates to be 0.022 min. The inactivation of 90% of the bacterial population at 62.5 °C prevented the formation of biofilm in the tested isolates [43].

The tested bacteria isolates tend to form a biofilm at 25 °C and 37 °C. However, some isolates may show poor adaptation to adhesion and biofilm formation. They did not produce a biofilm at the temperatures of 25 °C and 37 °C (*Pantoea* spp., *S. enterica* sv. *Typhimurium,* and *E. cloacae* no. 1) or produce a weak biofilm (*Klebsiella* spp. no. 2). Piras et al. (2015) showed that *S. enterica* strains, after overnight incubation, showed higher biofilm production at 22 °C than at 35.0 °C [35]. Stepanović et al. (2003) also observed higher biofilm production by *Salmonella* spp. at 30.0 °C after 24 h of incubation and at 22.0 °C after 48 h of incubation than at 37.0 °C [33]. In the study by Buzoleva et al. (2018), they showed that *Pantoea* spp. produces a more potent biofilm at 37 °C than at 22 °C but within 72 h of incubation [44]. Korres et al. (2013) showed that *K. pneumoniae* forms a stronger biofilm after 24 h at 40 °C than at 35.0 °C [45]. Hoštackǎ et al. (2010) compared the biofilm production by *K. pneumoniae* strains at 30.0 °C and 37.0 °C (24 h incubation). Two of the five strains showed a decrease in biofilm production at 37.0 °C. For the other three tested strains, the temperature of 30 °C was conducive to greater biofilm production [46]; this may indicate the presence of individual mechanisms of biofilm production in the same species of bacteria. A similar relationship was observed in our study, where the production of biofilm by five *Klebsiella* strains, two *E. hormachei* strains, and two *E. cloacae* strains was assessed. In each case, for the same strain, differences in biofilm production were observed depending on time, temperature, and inoculum count. *E. cloacae* no. 1 produced a very weak biofilm compared to *E. cloacae* no. 2. In the study of Nyenje et al. (2013), they showed that *E. cloacae* readily form biofilm on plastic surfaces. It was found that long incubation times and high temperatures affect the production of biofilm. At 37 °C, the bacteria produced a stronger biofilm after 24 and 48 h of incubation [47]. In addition, Iversen et al. (2004) showed that a higher temperature promotes biofilm formation (>40 °C) [48]. In a proprietary study at both 25 °C and 37 °C, *E. cloacae* produced a strong biofilm, except when it was incubated at 25 °C for 8 h; which may indicate that prolonged incubation times at lower temperatures will result in biofilm formation. In addition to the temperature and time factors, the study showed that the individual characteristics of the strains significantly affect the production of biofilm. The ability to produce biofilm by *E. hormaechei* bacteria was confirmed by the studies of Liu et al. (2022) [49]. The differences in biofilm production by *B. cereus* depending on environmental factors have been described in numerous studies. In the study by Kown et al. (2017), they showed that at 30.0 °C, increasing the incubation time from 24 to 48 h on a BHI medium increased biofilm production by two of the eight tested *B. cereus* strains [50]. Hayrapetyan et al. (2015) showed that approximately 50% of food-isolated *B. cereus* strains produced more biofilm after 24 h than after 48 h of incubation [51]. In our studies, *B. cereus* produced an equally strong biofilm in both 24 h and 48 h of incubation (at 37.0 °C). In our research, *C. sakazakii* did not show any signs of strong biofilm formation at 25.0 and 37.0 °C; moreover, the length of the incubation time had no effect on biofilm formation. Similar results were obtained by Ye et al. (2015), where strains of *C. sakazakii* formed only a weak biofilm at 37.0 °C [52]. The optimal temperature conditions for the development of a biofilm cannot be unequivocally determined. With some strains, lowering the incubation temperature may reduce bacterial growth but promote biofilm production. Moreover, the production of curli and cellulose usually takes place at temperatures below 30 °C [34].

PCA analysis made it possible to identify the structure of variables responsible for biofilm formation and their location in a limited space defined by two components. In the case of *C. sakazakii*, the first component explains more than 90% of the variability in the conditions of biofilm formation at 25 °C. Therefore, C1 can be called “temperature”. The temperature of 37 °C promotes the formation of biofilm by mesophilic bacteria. It can be seen that the variation in biofilm formation is time-dependent for both the S, M, and L inoculum count. The second component can therefore be called biofilm formation time. A similar interpretation of the main components can also be applied to *Bacillus* spp., *E. cloacae,* and *Salmonella* spp. However, in the case of *E. hormaechei* no. 2 and *Klebsiella* no. 3, the interpretation of the component factors is not so clear.

## 5. Conclusions

In this experiment, biofilm formation by bacteria isolated from breast milk was studied, which depended on factors such as: type and species of bacteria, temperature, inoculum count, and incubation time. Incubation temperatures assessed in this study (4.0, 25.0, 37.0, and 62.5 °C) are important for the life cycle of the tested bacterial strains and in the human milk logistics chain (from donor to beneficent). Studies on modeling surfaces show that pathogenic bacteria appearing in human milk as a result of cross-contamination tend to form a biofilm only at 25.0 and 37.0 °C. The phenomenon of biofilm formation by pathogenic bacteria isolates present in breast milk is significantly dependent on the temperature and individual mechanisms of biofilm production of the strains of the same kind or species of bacteria. The majority of isolates formed biofilm at both 25.0 and 37.0 °C after 12 h at an M inoculum count. Bacteria forming a strong biofilm at 25 and 37 °C were: *Bacillus* spp., *E. cloacae* strain 2, and *Klebsiella* spp. strains no. 1, no. 3, and no. 5.

Multivariate PCA showed that at 25.0 °C the biofilm formation by bacterial isolates was determined mainly by temperature; other factors such as inoculum size (three factors) and time (five factors) were less important. However, at 37.0 °C, time was the main factor in biofilm formation. From a safety point of view, it is extremely important to maintain proper hygiene and temperatures of rooms, surfaces, and equipment used for the pumping, storing, and preparation of human breast milk and PIF in institutions responsible for feeding newborns and premature babies. Performing hygienic activities can be supported by quality and safety assurance systems resulting from Codex Alimentarius, EU regulations, and ISO standards (9001:2016, 22 000:2018).

## Figures and Tables

**Figure 1 foods-11-03862-f001:**
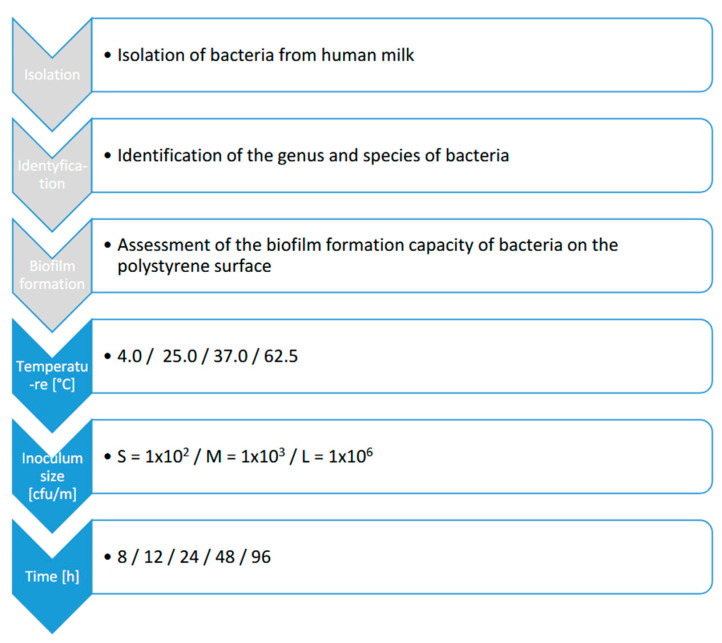
Research scheme.

**Figure 2 foods-11-03862-f002:**
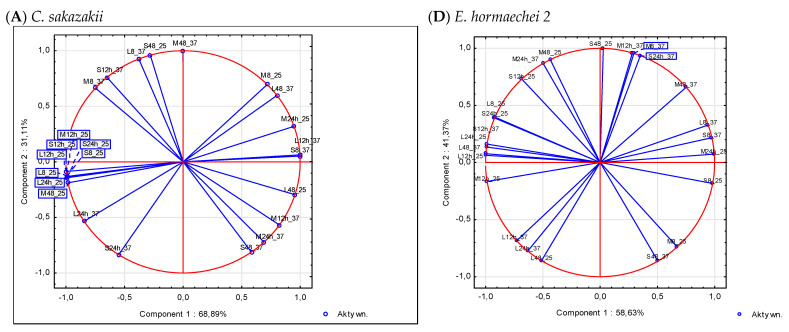
Principal component analysis of the conditions of biofilm formation on the surface by the tested bacterial isolates (**A**–**F**). S, M, and L: inoculum count; 8, 12, and 24: time of incubation [h].

**Table 1 foods-11-03862-t001:** Biofilm formation on the hydrophobic surface as a result of various combinations of inoculum size, time, and incubation temperature.

Isolate	Inoculum Count [CFU/mL]	Temperature 25.0 °C					Temperature 37.0 °C				
		Growth 48 h	Time of Biofilm Formation [h]				Growth 48 h	Time of Biofilm Formation [h]			
			8 h	12 h	24 h	48 h		8 h	12 h	24 h	48 h
*Cronobacter sakazakii* ATCC 25944	S	+	-	-	+	+	+	-	++	-	-
	M	+	-	+	++	+	+	-	++	-	+
	L	+	-	+	++	-	+	-	++	+	+
*Klebsiella* spp. 1	S	+	-	+++	++	++	+	+++	+++	+++	+++
	M	+	-	+++	++	-	+	-	+++	+++	++
	L	+	-	+	++	-	+	-	+++	+++	++
*Klebsiella* spp. 2	S	+	-	-	+	++	+	-	-	-	+
	M	+	-	-	-	-	+	+	-	+	-
	L	+	-	-	-	-	+	-	++	+++	+
*Klebsiella* spp 3	S	+	-	-	-	-	+	-	+++	++	++
	M	+	-	+	+++	-	+	+	+++	+++	++
	L	+	-	-	++	-	+	-	+	++	++
*Klebsiella* spp. 4	S	+	-	-	-	-	+	-	+++	+	+
	M	+	-	-	++	+	+	-	+++	++	+
	L	+	-	-	+	-	+	+++	++	+	+
*Klebsiella* spp. 5	S	+	-	-	+	++	+	-	++	++	+++
	M	+	-	+++	++	-	+	++	+++	+++	+++
	L	+	-	+	++	-	+	-	+++	+++	++
*Bacillus* spp.	S	+	-	+++	-	-	+	+++	+++	+++	+++
	M	+	-	+++	-	-	+	+++	+++	+++	+++
	L	+	-	+++	+	+	+	+++	+++	+++	+++
*Pantoea* spp.	S	+	-	-	-	-	+	-	-	-	+
	M	+	-	-	-	-	+	++	++	-	-
	L	+	-	-	-	-	+	-	-	-	+
*Salmonella enterica* subsp. Enterica sv. *Typhimurium*	S	+	-	-	-	-	+	+++	+++	+	+
	M	+	-	-	-	-	+	+	+++	+	+
	L	+	-	-	-	-	+	+	+	+	+
*Enterobacter hormaechei* 1	S	+	-	+	+	+	+	+	++	+	++
	M	+	-	++	++	++	+	+	+++	+	++
	L	+	-	++	+	-	+	++	++	+	++
*Enterobacter hormaechei* 2	S	+	-	+	-	-	+	+	+	+	++
	M	+	-	+	+	-	+	++	++	++	++
	L	+	-	+	+	-	+	++	+	+	+++
*Enterobacter cloacae* 1	S	+	-	-	-	-	+	-	-	-	-
	M	+	-	-	-	-	+	+	-	++	-
	L	+	-	-	-	-	+	-	-	-	-
*Enterobacter cloacae* 2	S	+	-	+++	+++	+++	+	+++	+++	+++	+++
	M	+	-	+++	+++	+++	+	+++	+++	+++	+++
	L	+	-	+++	+++	+++	+	+++	+++	+++	+++

- no biofilm; + weak; ++ moderate; +++ strong.

**Table 2 foods-11-03862-t002:** Quantitative biofilm formation by tested bacterial isolates and *C. sakazakii* [%], dependent on incubation time and inoculum count.

Temperature [°C]	Inoculum Count [CFU/mL]	Biofilm Classification	Time [h]			
			8	12	24	48
25.0	S	no	100%	58%	58%	58%
25.0		weak	0	17%	25%	8%
25.0		moderate	0	0%	8%	25%
25.0		strong	0	25%	8%	8%
25.0	M	no	100%	42%	42%	75%
25.0		weak	0	17%	8%	8%
25.0		moderate	0	8%	33%	8%
25.0		strong	0	33%	17%	8%
25.0	L	no	100%	50%	33%	83%
25.0		weak	0	25%	33%	8%
25.0		moderate	0	8%	25%	0
25.0		strong	0	17%	8%	8%
		Chi^2^	0.000000 *p* = 1.0000	2.285714 *p* = 0.89164	4.875000 *p* = 0.55994	4.038462 *p* = 0.67147
		R rank Spearman	-	0.0248283 *p* = 0.88571	0.1805556 *p* = 0.29198	−0.224370 *p* = 0.18832
37.0	S	no	50%	27%	25%	8%
37.0		weak	17%	9%	33%	33%
37.0		moderate	0	18%	17%	25%
37.0		strong	33%	45%	25%	33%
37.0	M	no	17%	17%	8%	25%
37.0		weak	42%	0%	25%	17%
37.0		moderate	25%	17%	25%	33%
37.0		strong	17%	67%	42%	25%
37.0	L	no	50%	17%	17%	8%
37.0		weak	8%	25%	33%	33%
37.0		moderate	17%	25%	8%	33%
37.0		strong	25%	33%	42%	25%
		Chi^2^	9.002381 *P* = 0.17344	5.421680 *p* = 0.049097	2.797203 *p* = 0.83384	2.781818 *p* = 0.83569
		R rank Spearman	−0.001714 *P* = 0.99208	−0.059207 *p* = 0.73548	0.1231952 *p* = 0.47410	−0.035722 *P* = 0.83614

**Table 3 foods-11-03862-t003:** Influence of temperature and time on biofilm formation by tested bacterial isolates.

		Chi^2^ and *p*-Value; *p* ≤ 0.05							
Temperature [°C]		Time [h]							
		8		12		24		48	
25.0	37.0	0.000000 *p* = 1.0000		218.36368 *p* = 0.03118		217.57780 *p* = 0.04040		211.80490 *p* = 0.22453	
Isolates		Temperature [°C]							
		25.0	37.0	25.0	37.0	25.0	37.0	25.0	37.0
		0.000000 *p* = 1.0000	50.80000 *p* = 0.02460	74.2857 *p* = 0.00005	53.97059 *p* = 0.01207	64.50000 *p* = 0.00084	67.91608 *p* = 0.00033	54.46154 *p* = 0.01076	79.85455 *p* = 0.00001

**Table 4 foods-11-03862-t004:** Determination coefficient values of R^2^ explaining the % variation in biofilm formation factors by principal components.

Variables	*C. sakazakii*		*Bacillus* spp.		*Klebsiella* spp. 3		*E. hormaechei* 2		*E. cloacae* 2		*Salmonella ent.* Typh.	
	C1	C2	C1	C2	C1	C2	C1	C2	C1	C2	C1	C2
S_8h_25	**0.98**	0.02	**0.98**	0.02	0.37	**0.63**	**0.97**	0.03	**0.48**	**0.52**	n.d.	n.d.
S_12h_25	**0.99**	0.01	**0.99**	0.01	**1.00**	0.00	0.47	**0.53**	**0.97**	0.03	n.d.	n.d.
**S_24h_25**	**0.98**	**0.02**	**0.98**	**0.02**	**0.99**	**0.01**	**0.85**	**0.15**	**0.18**	**0.82**	**n.d.**	**n.d.**
S_48h_25	0.08	**0.92**	0.08	**0.92**	0.04	**0.96**	0.00	**1.00**	0.01	**0.99**	n.d.	n.d.
M_8h_25	**0.52**	0.48	**0.52**	0.48	**0.52**	0.48	0.45	**0.55**	**0.80**	0.20	n.d.	n.d.
M_12h_25	**0.99**	0.01	**0.99**	0.01	**0.96**	0.04	**0.97**	0.03	**0.98**	0.02	n.d.	n.d.
M_24h_25	**0.90**	0.10	**0.90**	0.10	0.44	**0.56**	**0.99**	0.01	**1.00**	0.00	n.d.	n.d.
M_48h_25	**0.96**	0.04	**0.96**	0.04	0.10	**0.90**	0.19	**0.81**	0.13	**0.87**	n.d.	n.d.
L_8h_25	**0.98**	0.02	**0.98**	0.02	**1.00**	0.00	**0.84**	0.16	**0.68**	0.32	n.d.	n.d.
L_12h_25	**0.98**	0.02	**0.98**	0.02	**0.99**	0.01	**1.00**	0.00	**0.98**	0.02	n.d.	n.d.
L_24h_25	**0.98**	0.02	**0.98**	0.02	**0.99**	0.01	**0.97**	0.03	**0.98**	0.02	n.d.	n.d.
L_48h_25	**0.91**	0.09	**0.91**	0.09	0.02	**0.98**	0.26	**0.74**	**1.00**	0.00	n.d.	n.d.
S_8h_37	**1.00**	0.00	**1.00**	0.00	**0.63**	0.37	**0.95**	0.05	**0.87**	0.13	**0.81**	0.82
S_12h_37	0.42	**0.58**	0.42	**0.58**	**0.90**	0.10	**0.98**	0.02	**0.95**	0.05	**0.91**	1.00
S_24h_37	0.30	**0.70**	0.30	**0.70**	**0.83**	0.17	0.12	**0.88**	**0.84**	0.16	**0.68**	0.78
S_48h_37	0.34	**0.66**	0.34	**0.66**	0.15	**0.85**	0.25	**0.75**	**0.91**	0.09	**0.99**	1.00
M_8h_37	**0.56**	0.44	**0.56**	0.44	**0.99**	0.01	0.09	**0.91**	**0.97**	0.03	0.41	**0.95**
M_12h_37	**0.68**	0.32	**0.68**	0.32	**0.98**	0.02	0.08	**0.92**	**0.68**	0.32	**0.98**	0.98
M_24h_37	0.48	**0.52**	0.48	**0.52**	0.01	**0.99**	0.25	**0.75**	0.00	**1.00**	**0.91**	0.98
M_48h_37	0.00	**1.00**	0.00	**1.00**	0.07	**0.93**	**0.56**	0.44	0.39	**0.61**	**0.98**	1.00
L_8h_37	0.14	**0.86**	0.14	**0.86**	0.42	**0.58**	**0.89**	0.11	**1.00**	0.00	**0.72**	0.99
L_12h_37	**1.00**	0.00	**1.00**	0.00	0.00	**1.00**	**0.53**	0.47	**0.76**	0.24	**0.83**	1.00
L_24h_37	**0.71**	0.29	**0.71**	0.29	**0.98**	0.02	0.40	**0.60**	**0.98**	0.02	0.55	0.80
L_48h_37	**0.64**	0.36	**0.64**	0.36	0.45	**0.55**	**0.99**	0.01	0.19	**0.81**	**0.60**	0.99

n.d.—not detectable biofilm; C1: component 1; C2: component 2.

## Data Availability

Data is contained within the article.

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
