# Peer review of "Factors Influencing Biofilm Formation by Salmonella enterica sv. Typhimurium, E. cloacae, E. hormaechei, Pantoea spp., and Bacillus spp. Isolated from Human Milk Determined by PCA Analysis"

_foods, 2022, doi:10.3390/foods11233862_

Round 1

Reviewer 1 Report

The manuscript "Factors influencing biofilm formation by Salmonella enterica sv. Typhimurium, E. cloacae, E. hormaechei, Pantoea spp. and Bacillus spp. isolated from human milk determined by PCA analysis" has an interesting result. However, the manuscript needs to be improved. The discussion needs to review and compare the data with the literature. Many grammatically problematic sentences were found throughout the manuscript, which must be checked and corrected precisely.

1. L1-4: Bacteria names should be italic in the title.

2. L37: Reference required

3. L54: Use the full form of all abbreviated words for the first time throughout the manuscript.

4. L62: Bacteria names should be italic

5. L73: Authors are suggested to use updated references (last 5 years) in the introduction section.

6. L80: Bacteria names should be italic

7. L84: Use updated reference (https://doi.org/10.1016/j.foodres.2022.111163)

8. L103: Put reference (https://doi.org/10.3390/healthcare10030444)

9. L110: Use the full form of bacteria only when used for the first time throughout the manuscript.

10. L139: Check throughout the manuscript “CFU/mL”

11. L142-143: Re-write the sentence

12. L155: Avoid repetition. Delete “(Nest Biotechnology Co. China)”

13. L179, 227: P should be capitalized and italic “P ≤ 0.05”. Make corrections throughout the manuscript.

14. Many spacing and punctuation marks problems are found throughout the manuscript. Revision required.

Author Response

the answer is in the attachment

Reviewer 2 Report

Title: Factors influencing biofilm formation by Salmonella enterica sv. Typhimurium, E. cloacae, E. hormaechei, Pantoea spp. and Bacillus spp. isolated from human milk determined by PCA analysi

Manuscript Number:  foods-2035863

The manuscript entitled “Factors influencing biofilm formation by Salmonella enterica sv. Typhimurium, E. cloacae, E. hormaechei, Pantoea spp. and Bacillus spp. isolated from human milk determined by PCA analysi” is a more efficient research work on human milk. The author has investigated the mishandling of human milk and infant formula in infected premature and newborn babies. The author also studied the  effects of temperature, incubation time and initial cell concentration on biofilm formation. Most of the infections at 25°C and 37°C after 12h. The outcome of this work is to maintain suitable hygiene of the housing environment and feeding kits. The author wrote this article in good manner and a few English corrections need to do. I recommend the authors to undergo a thorough minor review of the manuscript for alignment corrections.

Abstract:

The author has to rewrite “Inappropriate hygiene during handling of human milk and powdered infant formula 15 (PIF) can result in infections in premature and newborn babies”

Line 109-110 : Why the author choose to collect 12 human milk samples? Why was the author not use healthy or controlled milk for this study? Please include hygiene maintained by mothers (handling, room hygiene, cleanness of baby items

Author Response

The answer is in the attachment
